DATA RELEASE

# Genome assembly of the roundjaw bonefish (*Albula glossodonta*), a vulnerable circumtropical sportfish

Brandon D. Pickett[1],[†], Sheena Talma[2],[†], Jessica R. Glass[3],[4], Daniel Ence[5], Timothy P. Johnson[6], Paul D. Cowley[3], Perry G. Ridge[1] and John S. K. Kauwe[1],[7],[*]

1  Department of Biology, Brigham Young University, Provo, Utah, USA
2  Department of Ichthyology and Fisheries Science, Rhodes University, Makhanda, South Africa
3  South African Institute for Aquatic Biodiversity, Makhanda, South Africa
4  College of Fisheries and Ocean Sciences, University of Alaska Fairbanks, Fairbanks, Alaska, USA
5  School of Forest Resources and Conservation, University of Florida, Gainesville, Florida, USA
6  Tim Johnson Gallery, Mesa, Arizona, USA
7  Brigham Young University – Hawai'i, Laie, Hawai'i, USA

## ABSTRACT

The roundjaw bonefish, *Albula glossodonta*, is the most widespread albulid in the Indo-Pacific and is vulnerable to extinction. We assembled the genome of a roundjaw bonefish from Hawai'i, USA, which will be instrumental for effective transboundary management and conservation when paired with population genomics datasets. The 1.05 gigabase pair (Gbp) contig-level assembly had a 4.75 megabase pair (Mbp) NG50 and a maximum contig length of 28.2 Mbp. Scaffolding yielded an LG50 of 20 and an NG50 of 14.49 Mbp, with the longest scaffold reaching 42.29 Mbp. The genome comprised 6.5% repetitive elements and was annotated with 28.3 K protein-coding genes. We then evaluated population genetic connectivity between six atolls in the Western Indian Ocean with 38,355 SNP loci across 66 *A. glossodonta* individuals. We discerned shallow population structure and observed genetic homogeneity between atolls in Seychelles and reduced gene flow between Seychelles and Mauritius. The South Equatorial Current might be the limiting mechanism of this reduced gene flow. The genome assembly will be useful for addressing taxonomic uncertainties of bonefishes globally.

**Subjects**  Genetics and Genomics, Evolutionary Biology, Marine Biology

**Submitted:**  22 July 2021

*  Corresponding author. E-mail:
kauwe@byu.edu

†  Contributed equally.

Preprint submitted at https://doi.org/10.1101/2021.09.10.458299

## DATA DESCRIPTION

Bonefishes (*Albula* spp.) are popular and economically important sportfishes found in the tropics around the globe. In the Florida Keys (Florida, USA) alone, $465 million of the annual economy is attributed to sportfishing tourism for bonefish and other fishery species inhabiting coastal flats [1]. Considering only bonefish, the sportfishing industry generates $169 million annually in the Bahamas [2, 3]. Unfortunately, population declines of bonefish have been observed around the globe, raising questions about how best to conserve bonefish and manage the associated fisheries [4]. *Albula* contains many morphological cryptic species, which, when combined with baseline data gaps, creates a considerable hurdle to effective management [5–7].

**Figure 1.** Roundjaw bonefish (*Albula glossodonta*) adult. Quantitative morphological data for this illustration of *A. glossodonta* were obtained primarily from Hidaka *et al.* [18] and Shaklee and Tamaru [14]. These were evaluated to select specific values for details such as the number of pored lateral line scales (76) and the number of rays in the pectoral (18), dorsal (16), pelvic (10), and anal fins (9). Each of these was portrayed in the illustration to be at or near the middle of the ranges reported in the reference articles. While some limited information was found in the literature describing coloration and general shape, the artist found particular benefit in some excellent, detailed photographs by Derek Olthuis of samples that Dr. J. S. K. Kauwe caught in Hawai'i and later genetically identified as *A. glossodonta*. A full-resolution version of this illustration can be viewed at https://www.timjohnsongallery.com/albula-glossodonta-illustration (also archived at the Internet Archive (https://web.archive.org) on 4 March 2022).

All bonefish species were historically synonymized to a single species, *Albula vulpes* [8], by 1940 [9–11], except for the threadfin bonefish, *A. nemoptera* [12], which is morphologically distinct [12, 13]. Molecular testing in the last several decades has enabled specific distinctions that were not previously possible [6, 9, 14–16]. Presently, three species complexes (*A. argentea*, *A. nemoptera*, and *A. vulpes* complexes) contain the 12 putative albulid species, although identification remains difficult in most cases [4]. The roundjaw bonefish, *A. glossodonta* (NCBI:txid121402; Fishbase ID: 11512) [17] (Figure 1), is one of seven species in the *A. vulpes* complex.

Most of the species in the *A. vulpes* complex can be found in the Caribbean Sea and Atlantic Ocean. By contrast, *A. glossodonta* can be found throughout the Indian and Pacific Oceans; this range overlaps slightly with *A. koreana* [19] from the *A. vulpes* complex and drastically with each species in the *A. argentea* complex [4]. *A. glossodonta* may be distinguished genetically from other species, but morphological identification based on its more rounded jaw and larger average size is difficult for non-experts [4, 20]. This difficulty, alongside underregulated fisheries and anthropogenic habitat loss, poses considerable threats to the future of this species. *A. glossodonta* has been evaluated as "Vulnerable" on the International Union for the Conservation of Nature's (IUCN) Red List of Threatened Species™ [7], and several incidents of overexploitation, including regional extirpation, have been reported [21–25].

## Context

The threat to *A. glossodonta* and other bonefish species will persist unless identification is made easier and population genomics techniques are employed to understand and identify evolutionarily significant units, areas of overlap between species, presence and extent of hybridization, and life-history traits, especially migration and spawning [4]. Genetic identification has previously been accomplished using only a portion of the mitochondrial cytochrome b gene and some microsatellite markers [6, 9, 15, 19, 26–33], which probably

**Table 1.** Sampling sites for *A. glossodonta* for population genomic analyses. The number of individuals (*N*) after data filtering are displayed for each atoll and island group.

| Island group | Atoll | *N* (Atoll) | *N* (Island group) |
|---|---|---|---|
| Amirantes | St. Joseph | 17 | 17 |
| Farquhar | Farquhar | 8 | 17 |
|  | Providence | 9 |  |
| Aldabra | Aldabra | 8 | 14 |
|  | Cosmoledo | 6 |  |
| Mauritius | St. Brandon | 18 | 18 |

provide an insufficient taxonomic history [4, 34–36]. To contribute to a more robust capacity for identification and enable more complex genomics-based analyses, we present a high-quality genome assembly of an *A. glossodonta* individual. A transcriptome assembly was also created and was used alongside computational annotation methods to create structural and functional annotations for the genome assembly. Additionally, we present results from a population genomic analysis of *A. glossodonta* populations in Seychelles and Mauritius, two island nations that support lucrative bonefish fly fishing industries.

## METHODS

An overview of the methods used in this study is provided here. Where appropriate, additional details, such as the code for custom scripts and the commands used to run software, are provided in the annotated shell scripts hosted in GigaDB [37].

## Tissue collection and preservation

Blood, gill, heart, and liver tissues from one *A. glossodonta* (NCBI:txid121402, Fishbase ID: 11512) individual were collected off the coast of Moloka'i (near Kaunakakai, Hawai'i, USA) in February 2016. Heart tissue from a second individual was also collected at the same location in September 2017 because of a failed sequencing run. Tissue samples were flash-frozen in liquid nitrogen, and blood samples were preserved in ethylenediaminetetraacetic acid (EDTA). All samples were packaged in dry ice for transportation to Brigham Young University (BYU; Provo, Utah, USA) and stored at −80 °C until sequencing. The blood sample from the first individual was used for short-read DNA sequencing. The gill, heart, and liver samples from the same individual were used for short-read RNA sequencing. The heart sample from the second individual was used for long-read sequencing and Hi-C sequencing.

For population genomic analyses, tissues (dorsal muscle samples or fin clips) were collected by fly fishing charter operators from 96 individuals of *A. glossodonta* from six coral atolls in the Southwest Indian Ocean (SWIO; Figure 1; Table 1). All tissues were preserved in 95% ethanol alcohol (EtOH) at −20 °C until sequencing, and thereafter cataloged and preserved in −80 °C in the tissue biobank of the South African Institute for Aquatic Biodiversity (Makhanda, South Africa).

## Sequencing
### DNA sequencing

DNA was prepared for long-read sequencing with Pacific Biosciences (PacBio; Menlo Park, California, USA) SMRTbell Library kits, following a manufacturer's protocol [38]. Continuous long-read (CLR) sequencing was performed on 13 SMRT cells for a 10-hour movie on the PacBio Sequel at the BYU DNA Sequencing Center (DNASC). Short-read

sequencing was performed in rapid run mode for 250 cycles in one lane on the Illumina (San Diego, California, USA) Hi-Seq 2500 at the DNASC after sonication with Covaris (Woburn, Massachusetts, USA) Adaptive Focus Acoustics technology and preparation with New England Biolabs (Ipswich, Massachusetts, USA) NEBNext Ultra II End Repair and Ligation kits with adapters from Integrated DNA Technologies (Coralville, Iowa, USA).

### mRNA sequencing

RNA was prepared with Roche (Basel, Switzerland) KAPA Stranded RNA-seq kit, following manufacturer recommendations. Paired-end sequencing was performed in high output mode for 125 cycles on the three samples together in one lane on the Illumina Hi-Seq 2500 at the DNASC.

### Hi-C sequencing

DNA was prepared with Phase Genomics (Seattle, Washington, USA) Proximo Hi-C kit (Animal) using the Sau3AI restriction enzyme (cut site: GATC) following recommended protocols. Paired-end sequencing was performed in rapid run mode for 250 cycles in one lane on the Illumina Hi-Seq 2500 at the DNASC.

### ddRAD library preparation and sequencing

Double digest restricted site-associated (ddRAD) sequencing was used to measure intraspecific genetic variation across six sampling localities in the SWIO. We extracted total DNA using Qiagen DNeasy Tissue kits according to the manufacturer's protocol (Qiagen, Inc., Valencia, California, USA). We visually examined the quality of DNA extractions using gel electrophoresis and by quantifying isolated DNA using a Qubit fluorometer (Life Technologies, Carlsbad, California, USA).

We modified a protocol developed by Peterson *et al.* [39] to prepare samples for ddRAD sequencing. We used the rare cutter *PstI* (5′-CTGCAG-3′ recognition site) and common cutter *MspI* (5′-CCGG-3′ recognition site). We carried out double digests of 150–200 ng total DNA per sample using the two enzymes in the manufacturer's supplied buffer (New England Biolabs) for 8 hours at 37 °C. We randomly distributed samples from different localities across the sequencing plate to minimize bias during library preparation. We visually examined samples using gel electrophoresis to determine digestion success, then ligated barcoded Illumina adapters to DNA fragments [39]. After ligation, we pooled samples into 12 libraries and performed a clean-up using the QIAquick PCR Purification kit. We then performed the polymerase chain reaction (PCR) using Phusion *Taq* (New England Biolabs) and Illumina-indexed primers [39]. Library DNA concentration was checked using a Qubit fluorometer, followed by normalization, a second round of pooling into four libraries, and an additional QIAquick cleanup step. We then re-measured DNA concentration using a Qubit and combined equal amounts from each of the four pools into one. We analyzed this final pool using a BioAnalyzer (Agilent, Santa Clara, California, USA) and performed size-selection using a Pippin Prep (Sage Science, Beverly, Massachusetts, USA), selecting for fragments between 300 and 500 bp. This was followed by a final measure of concentration using a BioAnalyzer. We sent the library to the University of Oregon Genomics and Cell Characterization Core Facility where concentrations were verified via quantitative PCR (qPCR) before 100-bp single-end sequencing on an Illumina Hi-Seq 4000.

### Read error correction

#### Illumina DNA

Illumina whole-genome sequencing (WGS) reads were corrected using Quake v0.3.5
(RRID:SCR_011839) [40], which depended upon old versions of R v3.4.0
(RRID:SCR_001905) [41] and the R package VGAM v0.7-8 [42, 43]. Quake attempts to
automatically choose a $k$-mer cutoff, traditionally based on $k$-mer counts provided by
Jellyfish (RRID:SCR_005491) [44]. To generate $q$-mer counts instead of $k$-mer counts,
BFCounter v0.2 (RRID:SCR_001248) [45] was used. Quake suggested a $q$-mer cutoff of 2.33,
which was subsequently used by the correction phase of Quake. Unlike the WGS reads, the
Illumina DNA reads created with the Hi-C library preparation were not corrected.

An estimate of the number of $k$-mers present in the reads is required to run BFCounter.
This number is based on the number of reads, the length of the reads, and $k$-mer size
according to this equation:

$$T = n(l - k + 1),$$

where $n$ is the number of reads, $l$ is the read length, $k$ is the $k$-mer size, and $T$ is the total
number of $k$-mers (not necessarily unique or distinct) present in the reads. This assumes a
uniform read length. If the reads are paired-end, $n$ is still the number of reads, not the
number of pairs of reads. Since ntCard v1.0.1 (RRID:SCR_022010) [46] was used to quickly
get a picture for the $k$-mer coverage histogram, its reported value F1 was used instead of the
equation as it is an estimate for $T$.

#### Illumina RNA

Illumina RNA-seq reads were corrected using Rcorrector v1.0.2 (RRID:SCR_022011) [47].
Rcorrector automatically chooses a $k$-mer cutoff based on $k$-mer counts provided by
Jellyfish [44], which Rcorrector runs automatically for the user. Alternately, Jellyfish can be
run externally or bypassed with an alternate $k$-mer counting program, and counts can
subsequently be provided to Rcorrector, which may be started at "stage 3". We bypassed
Jellyfish by using BFCounter v0.2 [45] to count $k$-mers. Note that Rcorrector made no
changes to the reads.

#### PacBio CLRs

Several methods were attempted to correct the PacBio CLRs. Corrected reads from each
method that did not fail were assembled, and the assembly results were used to choose the
correction strategy. Ultimately, a hybrid correction strategy was employed. Typically, a
"hybrid" correction strategy is one in which more than one data type (i.e., PacBio CLRs and
Illumina short reads) are employed. This differs from a "self" correction strategy in which
only the PacBio CLRs are used to correct themselves. Our hybrid strategy is not fully
described by the word "hybrid" because both a "self" and "hybrid" strategy were serially
employed; we referred to this strategy as a "dual" strategy (Figure 2). First, the reads were
self-corrected using Canu v1.6 (RRID:SCR_015880) [48]. Second, the self-corrected reads
were further corrected using Illumina short-reads (previously corrected with Quake) using
CoLoRMap (commit #baa680, RRID:SCR_022012) [49]. Note that Illumina short reads had to
be interleaved into a single file for CoLoRMap. Similarly, all PacBio reads were
concatenated into a single file, and the headers were required to be unique up to the first

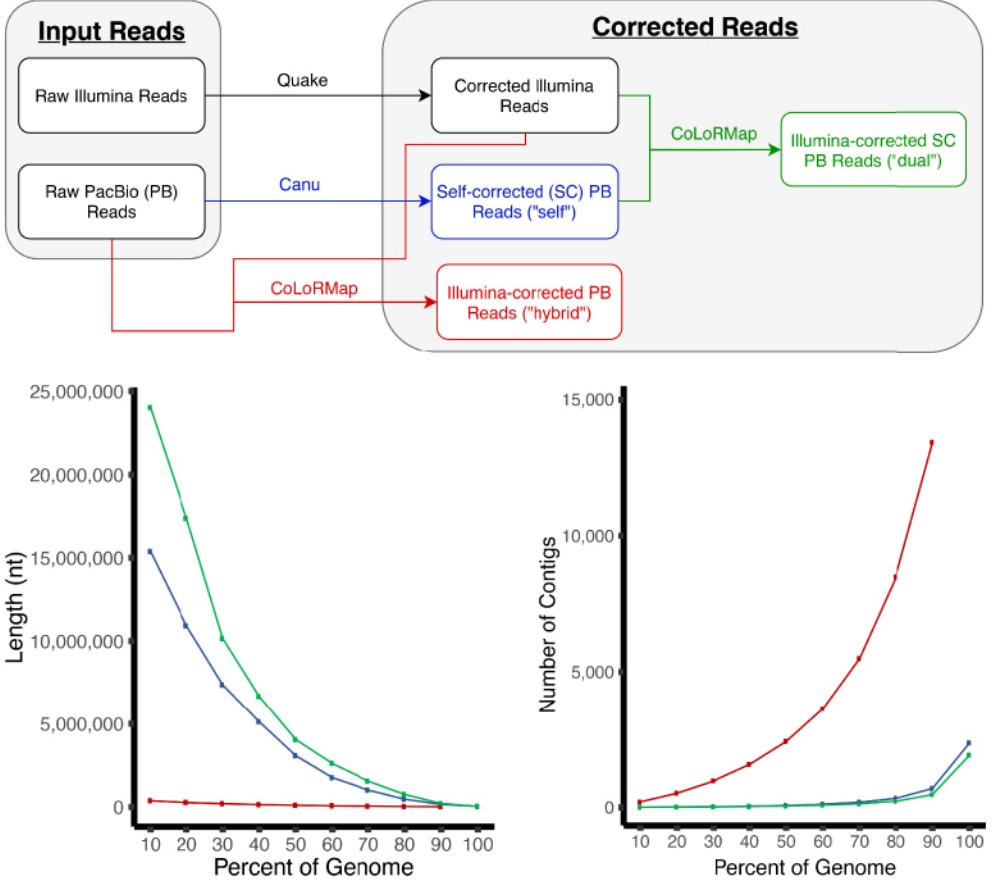

**Figure 2.** Long-read correction experiments. We explored the effects of several correction strategies on assembly continuity before settling on the chosen strategy (dual). Ignoring failed strategies owing to software failures, three strategies were employed: (blue) "self" correction (only Pacific Biosciences (PacBio) continuous long reads (CLRs)), (red) "hybrid" correction (using Illumina reads to correct the PacBio CLRs), and (green) "dual" correction (using Illumina reads to correct already self-corrected PacBio CLRs). Top panel: flow chart showing correction strategies. Bottom panel: NGx (left) and LGx (right) plots showing the contig-level assembly continuity per correction strategy. The x in NGx and LGx is a number between 0 and 100 representing the percentage of the genome size. NGx and LGx statistics are similar to Nx and Lx statistics except they are scaled to the genome size instead of the assembly size. Assemblies usually improve by maximizing and minimizing the areas under the NG and LG curves, respectively.

space. CoLoRMap is a pipeline with a basic wrapper script. In practice, it makes more sense to run each step in the wrapper script as separate jobs to avoid re-computing if a failure (e.g., too much random access memory (RAM) or time) occurs in a downstream step; this is how we ran the pipeline.

## Genome size estimation

Genome size was estimated using a *k*-mer analysis on the corrected Illumina WGS reads. First, the *k*-mer coverage was estimated using ntCard v1.0.1 [46]. The *k*-mer coverage histogram was computationally processed to calculate the area under the curve and identify the peak to determine genome size according to the following equation:

$$a/p = s,$$

**Table 2.** BUSCO statistics for the RNA transcripts and genomic assemblies.

| | Complete (%) | Complete single-copy (%) | Complete duplicated (%) | Fragmented (%) | Missing (%) | Total |
|---|---|---|---|---|---|---|
| **Transcriptome** | | | | | | |
| Trinity Transcripts | 3,144 (86.4) | 1,241 (34.1) | 1,903 (52.3) | 128 (3.5) | 368 (10.1) | 3,640 |
| **Genome** | | | | | | |
| Canu Contigs | 3,485 (95.7) | 3,081 (84.6) | 404 (11.1) | 22 (0.6) | 133 (3.7) | 3,640 |
| RaCon Polished contigs | 3,484 (95.7) | 3,076 (84.5) | 408 (11.2) | 22 (0.6) | 134 (3.7) | 3,640 |
| SALSA Scaffolds | 3,480 (95.6) | 3,074 (84.5) | 406 (11.2) | 27 (0.7) | 133 (3.7) | 3,640 |
| SALSA + Rascaf Scaffolds | 3,481 (95.6) | 3,076 (84.5) | 405 (11.1) | 25 (0.7) | 134 (3.7) | 3,640 |

where *a* is the area under the curve, *p* is the number of times the *k*-mers occur (the *x*-value) at the peak, and *s* is the genome size.

## Genome assembly, polishing, and scaffolding

Multiple assemblies were generated from various correction strategies. The final assembly was based on a hybrid correction strategy as described. The assembly was created using Canu v1.6 [48]. The assembly underwent two rounds of polishing with the corrected Illumina WGS reads using RaCon v1.3.1 (RRID:SCR_017642) [50]. Polished contigs were scaffolded in a stepwise fashion using two types of long-range information: Hi-C and RNA-seq reads. Both scaffolding steps required read-mapping to the contigs before determining how to order and orient contigs. The Hi-C data alignments were performed following the Arima Genomics (San Diego, California, USA) Mapping Pipeline [51], which relied on bwa v0.7.17-r1998 (RRID:SCR_010910) [52], Picard v2.19.2 (RRID:SCR_006525) [53], and SAMtools v1.6 (RRID:SCR_002105) [54]. BEDTools v2.28.0 (RRID:SCR_006646) [55] was used to prepare the Hi-C alignments for scaffolding. The RNA-seq data were aligned using HiSat v0.1.6-beta (RRID:SCR_015530) [56]. Scaffolding was performed for the Hi-C and RNA-seq data, respectively, with SALSA (commit #02018dc, RRID:SCR_022013) [57, 58], and Rascaf (RRID:SCR_022014) v1.0.2, commit #544ff4e [59]. Assembly continuity statistics, e.g., N50 and area under the NG-curve (auNG) [60, 61], were calculated with caln50 (commit #3e1b2be, RRID:SCR_022015) [62] and a custom Python script [37]. Assembly completeness was assessed using single-copy orthologs with Benchmarking Universal Single Copy Orthologs (BUSCO) v4.0.6 (RRID:SCR_015008) [63] and OrthoDB v10 (RRID:SCR_011980) [64] (Table 2).

## Transcriptome assembly

The transcriptome was assembled from Illumina RNA-seq reads from all three tissues (i.e., gill, heart, and liver). The raw reads were used because Rcorrector did not modify any bases, thus making the raw reads and the "corrected" reads identical. The transcripts were assembled using Trinity v2.6.6 (RRID:SCR_013048) [65], which depended on Bowtie v2.3.4.3 (RRID:SCR_016368) [66], Jellyfish v2.2.10 (RRID:SCR_005491) [44], salmon v0.12 (RRID:SCR_017036) [67], and SAMtools v1.6 [54]. Assembly completeness was assessed using single-copy orthologs with BUSCO v4.0.6 [63] and OrthoDB v10 [64] (Table 2).



**Table 3.** Input parameters for *ipyrad* used to assemble ddRAD data to the *A. glossodonta* reference genome.

| Parameter | Description | Input |
|---|---|---|
| assembly_method | Assembly method | reference |
| datatype | Datatype | ddrad |
| restriction_overhang | Restriction overhang (cut1,) or (cut1, cut2) | TGCAG, CCG |
| max_low_qual_bases | Max low quality base calls (Q < 20) in a read | 5 |
| phred_Qscore_offset | phred Q score offset | 33 |
| mindepth_statistical | Min depth for statistical base calling | 6 |
| mindepth_majrule | Min depth for majority-rule base calling | 6 |
| maxdepth | Max cluster depth within samples | 10,000 |
| clust_threshold | Clustering threshold for de novo assembly | 0.9 |
| max_barcode_mismatch | Max number of allowable mismatches in barcodes | 0 |
| filter_adapters | Filter for adapters/primers | 2 |
| filter_min_trim_len | Min length of reads after adapter trim | 35 |
| max_alleles_consens | Max alleles per site in consensus sequences | 2 |
| max_Ns_consens | Max N's (uncalled bases) in consensus | 0.05 |
| max_Hs_consens | Max Hs (heterozygotes) in consensus | 0.05 |
| min_samples_locus | Min number of samples per locus for output | 10 |
| max_SNPs_locus | Max number of SNPs per locus | 0.2 |
| max_Indels_locus | Max number of of indels per locus | 8 |
| max_shared_Hs_locus | Max number of heterozygous sites per locus | 0.5 |
| trim_reads | Trim raw read edges (R1>, <R1, R2>, <R2) | 0, 0, 0, 0 |
| trim_loci | Trim locus edges (R1>, <R1, R2>, <R2) | 0, 0, 0, 0 |

## ddRAD sequence assembly and filtering

We assembled all ddRAD data using the program ipyrad v0.9.31 (RRID:SCR_022016) [68]. The input parameters for ipyrad are shown in Table 3. All *A. glossodonta* reads were mapped to the genome assembly.

The seven-step ipyrad workflow was as follows:

(1) Sequences were demultiplexed by identifying individual sample barcode sequences and restriction overhangs.

(2) Barcodes and adapters were trimmed from reads, which were then hard-masked using a *q*-score threshold of 20 and filtered for a maximum number of undetermined bases per read.

(3) Reads were clustered with a minimum depth of coverage of six to retain clusters in the ddRAD assembly.

(4) Sequencing error rate and heterozygosity were jointly estimated from site patterns across the clustered reads assuming a maximum of two consensus alleles per individual.

(5) Consensus base calls were determined for each allele using the parameters from step four and filtered for a maximum number of undetermined sites per locus.

(6) Consensus sequences and aligned reads were clustered for each sample.

(7) Data were filtered by the maximum number of alleles per locus, the maximum number of shared heterozygous sites per locus, and other criteria [68], and output files were formatted for downstream analyses. We included all loci shared by at least 10 individuals.

We performed additional filtering steps after running ipyrad to account for missing data and rare alleles. Using VCFtools v0.1.16 (RRID:SCR_001235) [69] and BCFtools v1.6 (RRID:SCR_002105) [54], we removed individuals missing more than 98% of genotype calls. We retained only biallelic single nucleotide polymorphisms (SNPs) and removed (i) indels,



**Table 4.** Data filtering steps implemented in VCFtools and PLINK after assembly in ipyrad.

| SNP quality filters | |
|---|---|
| Genotype calls | Remove individuals missing >98% genotype calls |
| Indels | Remove indels |
| Read depth | Remove loci with mean depth >100 |
| Singletons and minor alleles | Retain sites with a minor allele frequency > 0.05 and minor allele count ≥2 |
| Biallelic SNPs | Max alleles = 2 |
| **Missing data** | |
| | Remove loci with genotype call rate <40% |
| | Remove individuals missing >60% genotype calls |
| | Remove loci with genotype call rate <60% |
| | Remove individuals missing >50% genotype calls |
| | Remove loci with genotype call rate <75% |
| **Hardy–Weinberg Equilibrium (HWE)** | |
| Hardy–Weinberg Equilibrium | Remove loci out of HWE (0.05) |
| **Linkage disequilibrium** | |
| Linkage disequilibrium | Remove loci within 1-Kbp windows with $r^2$ > 0.6 |

(ii) loci with minor allele frequencies <0.05 to exclude singletons and false polymorphic loci due to potential sequencing errors, (iii) alleles with a minimum count <2, and (iv) loci with high mean depth values (>100). We then implemented an iterative series of filtering steps based on missing data and genotype call rates to maximize genomic coverage per individual (Table 4) [70]. Thereafter, we removed loci out of Hardy–Weinberg Equilibrium to filter for excess heterozygosity. We then used PLINK v1.9 (RRID:SCR_001757) [71] to perform linkage disequilibrium pruning by calculating the squared coefficient of correlation ($r^2$) on all SNPs within a 1-kilobase pair (Kbp) window [72]. We removed all SNPs with an $r^2$ value greater than 0.6.

## Computational annotation of assembled genome

The MAKER v3.01.02-beta (RRID:SCR_005309) [73] pipeline was used to annotate the assembly. With minor modifications (see Figure 3 and protocols.io [74]), annotation proceeded as described in the Maker Wiki tutorial [75]. A custom repeat library was created using RepeatModeler v1.0.11 (RRID:SCR_015027) [76]. The transcriptome assembly, genome assembly, and proteins from UniProtKB Swiss-Prot (RRID:SCR_002380) [77, 78] were used as input to MAKER to create initial annotations. Gene models based on these annotations were used to train the following *ab initio* gene predictors: AUGUSTUS v3.3.2 (RRID:SCR_008417) [79, 80] and SNAP (commit #daf76ba, RRID:SCR_005501) [81]. AUGUSTUS was trained using BUSCO [63] as a wrapper; SNAP was trained without a wrapper. Genemark-ES v4.38 (RRID:SCR_011930) [82–84] was also trained on the assembled genome. These models were all provided to MAKER for a second round of structural annotation, which utilized EVidenceModeler v1.1.1 (RRID:SCR_014659) [85]. The gene models based on those annotations were filtered with gFACs v1.1.1 (RRID:SCR_022017) [86] and again provided to AUGUSTUS and SNAP. As Genemark-ES does not accept initial gene models, it did not need to be run again. The gene models from the *ab initio* gene predictors were again provided to MAKER for a third and final round of annotation, in which tRNAs were searched for with tRNAscan-SE v1.3.1 (RRID:SCR_010835) [87]. Functional annotations were added using MAKER accessory scripts, the BLAST+ Suite v2.9.0 (RRID:SCR_004870) [88, 89], and InterProScan v5.45-80.0 (RRID:SCR_005829) [90, 91]. The annotations in GFF3 format were validated with GenomeTools v1.6.1 (RRID:SCR_016120) [92] and manually curated to

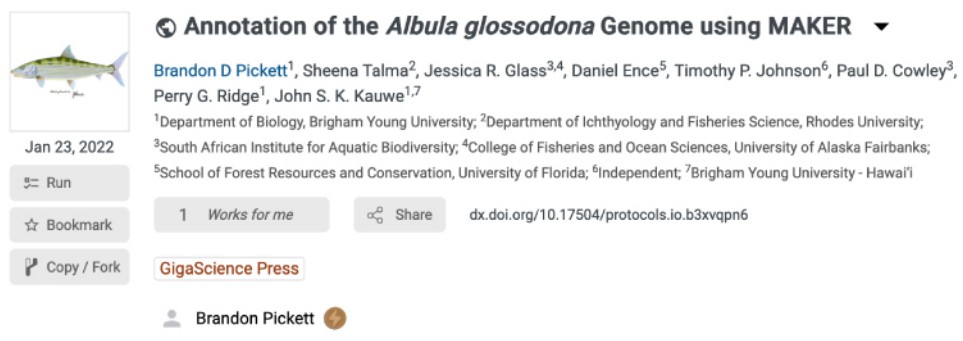

**Figure 3.** Protocols.io protocol for annotation of the *Albula glossodonta* genome using MAKER [3]. https://www.protocols.io/widgets/doi?uri=dx.doi.org/10.17504/protocols.io.b3xvqpn6

adhere to GenBank submission guidelines. Repeat characterization was performed with RepeatMasker v4.1.2-p1 (RRID:SCR_012954) [93] using Dfam v3.3 (RRID:SCR_021168) [94] and the RepBase RepeatMasker Library v20181026 (RRID:SCR_021169) [95, 96].

## Statistical analysis of population genomic data
### *Detection of loci under selection*
Before conducting population genomic analyses, we performed outlier tests to identify loci putatively under selection. These are usually identified by a significant difference in allele frequencies between populations [97]. Specifically, we implemented two outlier detection methods that accommodate missing data: pcadapt v4.1.0 (RRID:SCR_022019) [97] and BayeScan v2.1 (RRID:SCR_022018) [98]. The assumption behind pcadapt is that loci associated with population structure, ascertained via principal component analysis (PCA), are under selection [97]. pcadapt is advantageously fast and able to handle many loci. The number of principal components (*K*) was chosen based on visualization of a scree plot of the eigenvalues of a covariance matrix. Once the *K*-value was chosen, the Mahalanobis distance (*D* test statistic) was calculated using multiple linear regression of the number of SNPs versus *K* [97, 99]. To account for false discovery rates, the *p*-values generated using the Mahalanobis distance *D* were transformed to *q*-values using the R v3.6.3 [41] *q*-value package v2.15.0 (RRID:SCR_001073) [100] with the cut-off point (*a*) set to 10% (0.1).

BayeScan measures allele frequencies between different populations and identifies loci perceived to be undergoing natural selection based on their $F_{ST}$ values [101, 102]. The method applies linear regression to generate population- and locus-specific $F_{ST}$ estimates and calculates subpopulation $F_{ST}$ coefficients by taking the difference in allele frequency between each population and the common gene pool. BayeScan incorporates uncertainties in allele frequencies owing to small sample sizes, as well as imbalances in the number of samples between populations [98]. We assigned each of the six sampling localities as a population. Our analyses were based on 1:50 prior odds and included 100,000 iterations and a false discovery rate of 10%. We used the default values for the remaining parameters and visualized results in R v3.6.3 following the developer's manual [103]. After running both pcadapt and BayeScan, we used R to assess the number of outliers identified by both programs and subsequently removed outlier loci to generate a neutral dataset for downstream analyses. The outlier dataset results are presented in Figure 4. These datasets

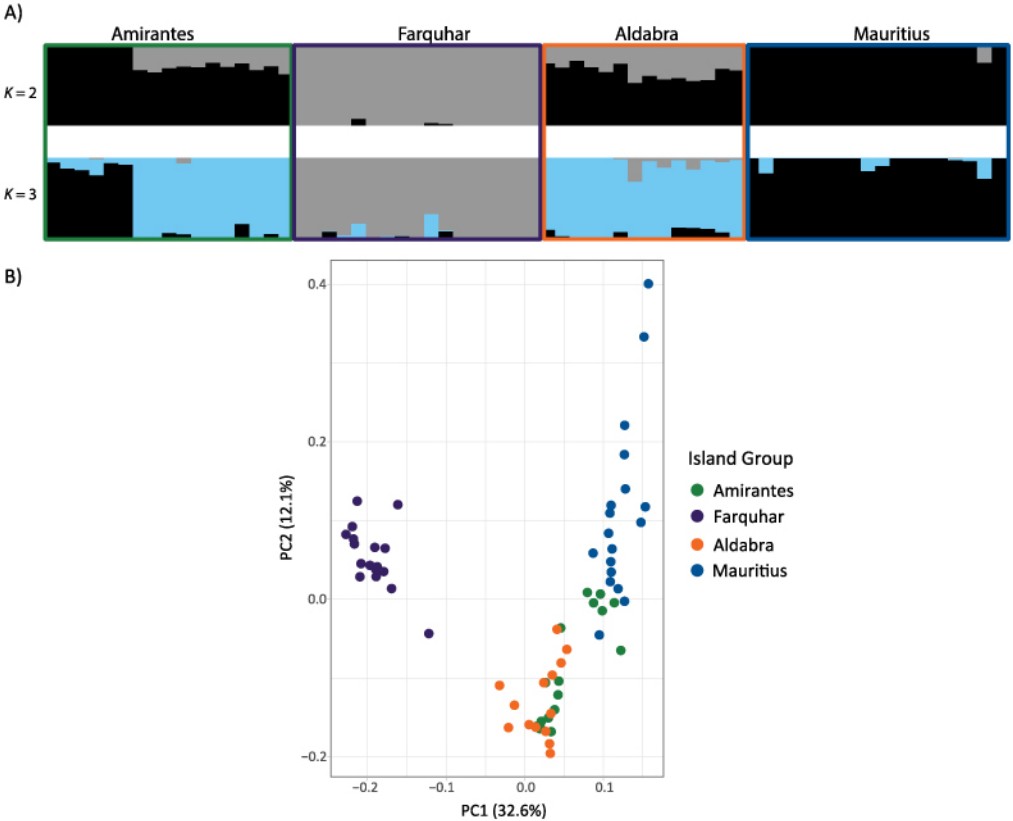

**Figure 4.** Population differentiation analyses for the outlier single nucleotide polymorphism (SNP) loci. We analyzed 155 outlier SNP loci for population structure. Cross-entropy scores generated by sparse non-negative matrix factorization (sNMF) gave strongest support for *K* = 3 populations of *Albula glossodonta*. Evidence of genetic differentiation between island groups was displayed in individual ancestry plots generated in sNMF (A), as well as principal component analysis (PCA) biplots of the first two principal components (B). Both individual ancestry plots and PCA results indicated differentiation in the outlier loci dataset was strongest between Farquhar and the other island groups.

were also annotated with respect to the MAKER-based annotations of the genome assembly using SnpEff v5.0e (RRID:SCR_005191) [104].

### Population structure and genetic differentiation

To examine population structure, we used a model-based clustering method to reconstruct the genetic ancestry of individuals using sparse non-negative matrix factorization (sNMF) and least-squares optimization. Model-based analyses were performed using the package LEA v2.6.0 (RRID:SCR_022020) [105] in R. The sNMF function in LEA estimates the number of ancestral populations and the probability of the number of gene pools from which each individual originated by calculating an ancestry coefficient and investigating the model's fit through cross-entropy criterion [106]. We calculated and visualized cross-entropy scores of *K* population clusters ranging from 1–10 with 10 replicates. To complement sNMF, we also used PCA, a distance-based approach based on variation in allele distributions, implemented in VCFtools v0.1.16 [69]. For sNMF and PCA analyses, no populations were assigned *a priori*. We assigned each of the six sampling localities as populations for subsequent visualization, grouped into four "island groups" based on the proximity of some

**Table 5.** Sequencing information.

| Company | Illumina | Illumina | Illumina | PacBio |
|---|---|---|---|---|
| **Instrument** | Hi-Seq 2500 | Hi-Seq 2500 | Hi-Seq 2500 | Sequel I |
| **Mode** | Rapid run | High output | Rapid run | N/A |
| **Sequencing type** | PE | PE | Hi-C, PE | SMRT, CLR |
| **Duration** | 250 cycles | 125 cycles | 250 cycles | 30 hours |
| **Specimen** | 1 | 1 | 2 | 2 |
| **Tissues** | Blood | Gill, Heart, Liver | Heart | Heart |
| **Molecule** | DNA | RNA | DNA | DNA |
| **Millions of read(pair)s** | 109.5 | 270.7 | 88.7 | 9.5 |
| **Mean read length (bp)** | 246 | 124 | 245 | 7,353 |
| **Read N50 (bp)** | 250 | 125 | 250 | 13,831 |
| **Nucleotides (Gbp)** | 53.9 | 67.2 | 44.3 | 69.9 |

The results from each type of DNA and RNA sequencing from *Albula glossodonta*. bp: basepairs; CLR: continuous long-reads; Gbp: gigabase pair; PacBio: Pacific Biosciences; PE: Paired-end reads; SMRT: Single-Molecule, Real-Time sequencing.

of the atolls that comprised our sampling localities (Figure 5). The five Seychelles atolls we sampled were spread among three separate clusters of islands, which are commonly referred to as the "outer" island groups owing to the geographic locations of these outlying coralline islands relative to the densely populated, granitic "inner" islands of the Seychelles archipelago. The island groups consisted of (i) Amirantes (St. Joseph's Atoll), (ii) Farquhar (Farquhar and Providence Atolls), (iii) Aldabra (Aldabra and Cosmoledo Atolls), as well as (iv) Mauritius (St. Brandon's Atoll; Table 1). We computed summary statistics in R v3.6.3, including pairwise $F_{ST}$ estimates (StAMPP v1.6.1 (RRID:SCR_022022) [107]), isolation by distance via the Mantel Rand test (adegenet v2.1.3 (RRID:SCR_000825) [108]), and expected and observed heterozygosity (hierfstat v0.5-7 (RRID:SCR_022021) [109]) to compare genetic diversity and differentiation between the four island groups.

## DATA VALIDATION AND QUALITY CONTROL
### Sequencing
#### DNA sequencing
Paired-end, short-read sequencing (Illumina) yielded 109.5 million pairs of reads comprising 53.86 Gbp. The mean and N50 read lengths were 245.981 and 250, respectively. Continuous long-read sequencing (PacBio) generated 9.5 million reads with a total of 69.85 Gbp. The mean and N50 read lengths were 7,352.726 and 13,831, respectively. The longest read was 103,889 bp. The read length distribution is plotted in Figure 5. Result summaries for both sequencing runs are available in Table 5.

#### mRNA sequencing
RNA-seq from the three tissues (i.e., gill, heart, and liver) generated 270.7 million pairs of reads totaling 67.2 Gbp. The gill tissue yielded 107.7 million pairs of reads, with a total of 26.7 Gbp. The heart tissue generated 19.6 Gbp across 78.8 million pairs of reads. The 84.2 million pairs of reads from the liver tissue comprised 20.9 Gbp. Across all three tissues, the mean and N50 read lengths were 124.122 and 125, respectively. The combined results from all three tissues are summarized in Table 5.

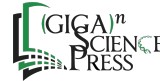

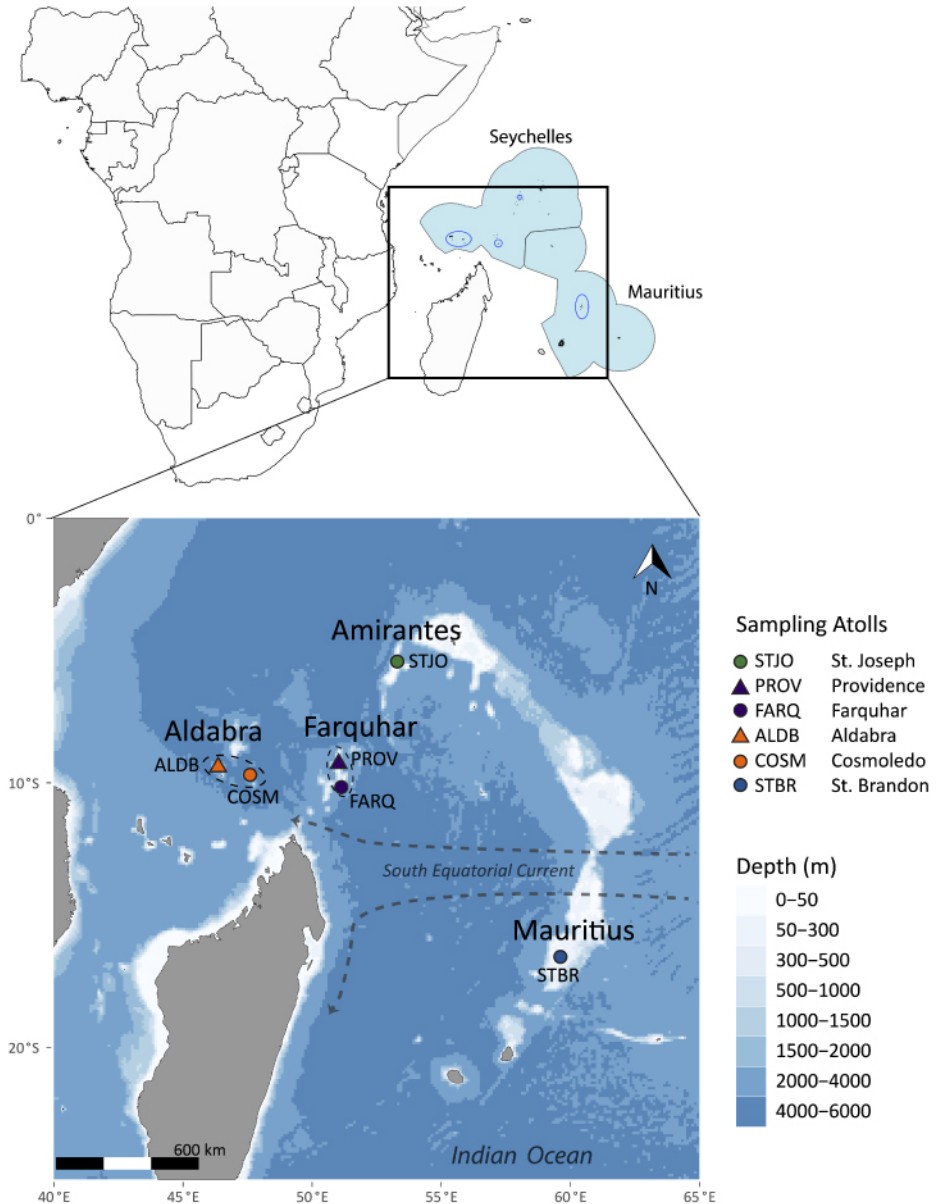

**Figure 5.** Sampling localities for *Albula glossodonta* population genomic analysis. The upper panel shows the marine boundaries for the Seychelles and Mauritius in light blue. Locations of sampling sites are indicated by dark blue ovals. The lower panel shows the atolls comprising the four island groups: Amirantes, Farquhar, Aldabra, and Mauritius.

### Hi-C sequencing

Sequencing yielded 88.7 million pairs of reads comprising 44.28 Gbp. The mean and N50 read lengths were 249.493 and 250, respectively. A summary of these results is presented in Table 5.



### ddRAD sequencing

After data processing using ipyrad, we recovered a mean of 114,324 reads per individual and an average of 107,105 loci per individual. Following filtering for missing data, minor allele frequency, and linkage disequilibrium, the dataset contained 66 individuals and 38,355 SNPs. BayeScan, being a more conservative outlier detection method than pcadapt, did not identify any outliers; we thus used only outlier detection results from pcadapt. Subsequent removal of pcadapt outliers ($N$ = 155) resulted in a neutral dataset containing 38,200 SNPs with 9% missing data.

## Read error correction

### Illumina DNA

When Quake corrects paired-end reads, three outcomes are possible for each pair of reads: (i) both reads are either already correct or correctable, (ii) one read is either correct or correctable and the other is low-quality, or (iii) both reads are low-quality. Of the original 218.96 million reads (109.5 million pairs of reads), Quake corrected 62.7 million of them and removed 51.6 million of them. 5.97 million pairs of reads were discarded because both reads were rated as erroneous. 39.6 million pairs of reads had one read that was correct or correctable and one read that was low-quality; these were also discarded. The remaining 63.88 million pairs of reads were either correct or correctable and were kept in the final read set containing 29.11 Gbp of sequence.

### Illumina RNA

No corrections were made to the RNA-seq reads by Rcorrector [47].

### PacBio CLRs

The dual-correction strategy (self-correction followed by hybrid-correction) reduced the number of reads from 9.5 million to 2.79 million and the total number of bases from 69.85 Gbp to 36.79 Gbp. The mean and N50 read lengths were changed from 7,354 and 13,831 to 13,193 and 15,483, respectively. The longest read was 63,271 bases. The distribution of read lengths can be viewed in Figure 6.

## Genome size estimation

The genome size was estimated to be approximately 0.933 Gbp as a result of the *k*-mer analysis, which was consistent with the authors' expectations based on two closely related elopomorph species [110, 111].

## Genome assembly, polishing, and scaffolding

The initial assembly from Canu comprised 3,799 contigs with a total assembly size of 1.05 Gbp. The mean contig length, N50, NG50 [112], and maximum contig length were 276.2 Kbp, 3.6 Mbp, 4.7 Mbp, and 28.2 Mbp, respectively. The L50 was 57, and the LG50 was 43. The auNG was 8.17 million. After two rounds of polishing these contigs with the corrected Illumina WGS reads using RaCon, the assembly statistics changed only marginally. The number of contigs, L50, and LG50 were unchanged. The assembly size decreased by 318.7 Kbp (0.03%). The mean contig length, N50, NG50, and maximum contig length were reduced by 83.8 bp (0.03%), 1.3 Kbp (0.04%), 1.5 Kbp (0.03%), and 3.8 Kbp (0.01%), respectively. The auNG decreased by 2 Kbp (0.02%).

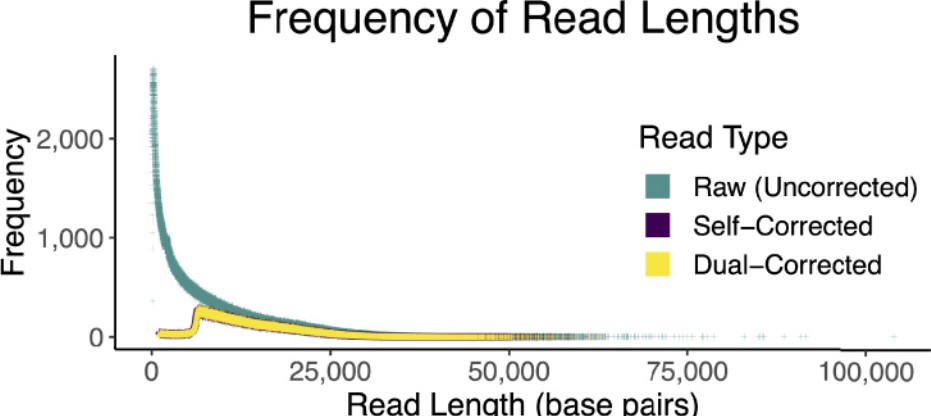

**Figure 6.** Read length frequencies of Pacific Biosciences continuous long reads (PacBio CLRs). The change in read length distribution is demonstrated as reads are corrected. The dramatic shift from raw to corrected reads is evident. The self-corrected (purple) data points are slightly larger than the dual-corrected (yellow) data points to make the purple distribution visible, the size has no meaning.

The scaffolding with the Hi-C data joined some polished contigs together, reducing the sequence count to 3.6K (−4.69%). The number of bases, excluding unknown bases (Ns), was unchanged; however, it is important to note that when SALSA creates gaps while ordering and orienting contigs, it always uses a gap size of 500 bp. The result, in this case, was adding 116 Kbp of Ns, which means 232 gaps were created. These gaps were spread across 113 scaffolds. No scaffold had more than six gaps (seven contigs ordered and oriented together). The mean scaffold length, scaffold N50, scaffold NG50, and maximum scaffold length increased by 13.6 Kbp (4.92%), 3.8 Mbp (106.25%), 5.79 Mbp (121.90%), and 14.1 Mbp (49.85%), respectively. Coupled with these increases were decreases of 29 (50.88%) and 22 (51.16%) in the L50 and LG50, respectively. The auNG increased to 14.1 million (+72.81%). The quality of the Hi-C scaffolding can be visualized (Figure 7) via a contact matrix generated by PretextMap (RRID:SCR_022023) [113] and PretextView (RRID:SCR_022024) [114].

The genome assembly was further improved by scaffolding with RNA-seq data. As expected, the magnitude of the changes between sets of scaffolds was smaller than what was observed between contigs and scaffolds. The total number of sequences was reduced by 176 to 3.4K (−4.69%). The number of known bases was again unchanged; however, it is important to note that when Rascaf orders and orients contigs (or other scaffolds) it always inserts a gap of 17 bp to represent gaps of unknown size. Rascaf added 179 new gaps (3,043 unknown bases) across 148 sequences. Three gaps (1,500 unknown bases) from SALSA were removed, but the rest remained unchanged. The most gaps added to a single sequence by Rascaf was five. The sequence with the most total gaps (from either source) had seven gaps (six from Hi-C), thus eight contigs were joined together.

This resulting set of scaffolds (which also includes all the contigs that were not joined to another contig in some way) had a mean length of 304.5 Kbp (+5.11% from the Hi-C only value) and a maximum length of 42.29 Mbp (+0.08%). The N50 and NG50 increased to 7.97 Mbp (+7.04%) and 14.49 Mbp (+37.58%), respectively. Decreases to 26 (−7.14%) and 20 (−4.76%) were observed for L50 and LG50, respectively. The auNG increased to 14.7 M

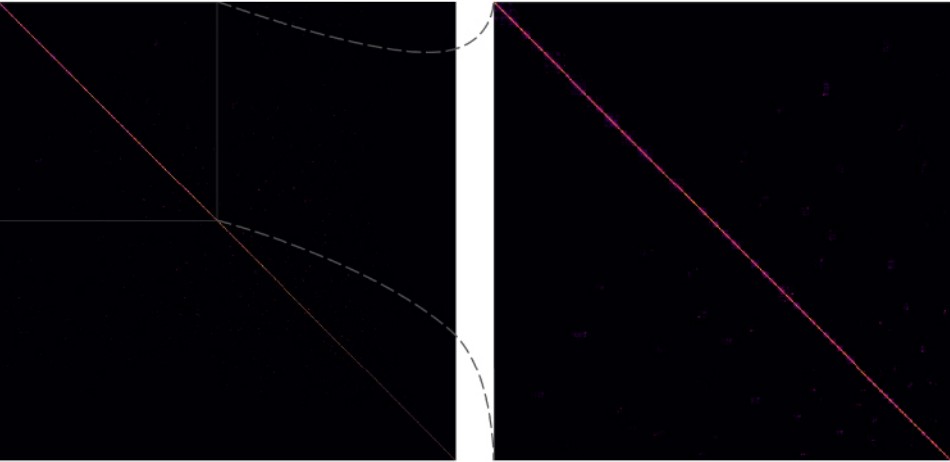

**Figure 7.** Hi-C contact matrix showing scaffolding correctness. In the context of scaffolding, Hi-C contact matrices show how correct the scaffolds are. Off-diagonal marks, especially those that are bright and large, are evidence of mis-assembly and/or incorrect scaffolding. The interpretations of the lighter and smaller off-diagonal marks in this image are ambiguous because the assembly is unphased with some relatively short contigs/scaffolds. Additional detail may be viewed by zooming in on the high-resolution image.

**Table 6.** Continuity statistics.

|  | Contigs | Polished Contigs | Scaffolds (Hi-C) | Scaffolds (Hi-C + RNA-seq) |
|---|---|---|---|---|
| Sequences | 3,799 | 3,799 | 3,621 | 3,445 |
| Known bases (Gbp) | 1.04935 | 1.04903 | 1.04903 | 1.04903 |
| Mean length | 276,217.073 | 276,133.196 | 289,707.267 | 304,507.986 |
| Max. length | 28,203,290 | 28,199,443 | 42,256,846 | 42,290,388 |
| N50 | 3,612,671 | 3,611,341 | 7,448,473 | 7,972,764 |
| NG50 | 4,747,926 | 4,746,442 | 10,532,420 | 14,490,288 |
| NG90 | 503,090 | 503,135 | 739,806 | 827,489 |
| LG50 | 43 | 43 | 21 | 20 |
| LG90 | 289 | 289 | 181 | 162 |
| auNG | 8,165,188 | 8,163,173 | 14,106,761 | 14,723,001 |
| Sequences with gaps | – | – | 133 | 236 |
| Gaps | – | – | 232 | 408 |
| Unknown bases | – | – | 116,000 | 117,543 |
| Mean gap length | – | – | 500.000 | 288.096 |

Continuity statistics for the *Albula glossodonta* genome assembly at various stages. The "Scaffolds (Hi-C + RNA-seq)" column represents the final assembly. Also note that when submitted to GenBank, the gaps were all converted to a length of 100 bp. Unless otherwise specified, all nucleotide sequences are measured in base pairs (bp).

(+4.37%). Table 6 summarizes the assembly continuity statistics, and the auNG is visualized in Figure 8.

The assembly completeness, as assessed with single-copy orthologs, was also evaluated at each stage (Table 2). The results suggest that the modifications made to the primary Canu-based assembly from polishing and scaffolding did not significantly impact the correct assembly of single-copy orthologs. The final set of scaffolds had 3,481 complete single-copy orthologs (95.6% of 3,640 from the ODB10 *Actinopterygii* set). Of these 88.4% (3,076) were present in the assembly only once, and 11.6% (405) were present more than once. Twenty-five (0.7%) and 135 (3.7%) single-copy orthologs were fragmented in and missing from the assembly, respectively.

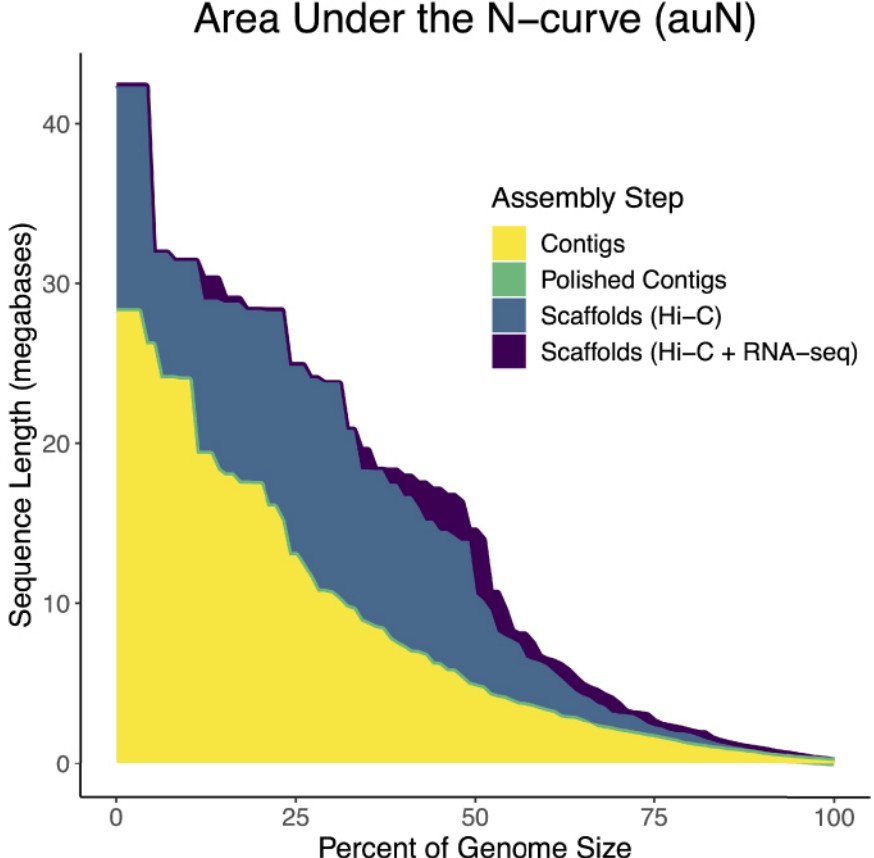

**Figure 8.** Area under the NG curve (auNG) for each assembly step. The NG-curve and the area under it are plotted for each major step of the assembly: contigs, polished contigs, scaffolds from only Hi-C data, and scaffolds from both Hi-C and RNA-seq data. The auNG for the polished contigs (green) is very similar to the contigs (yellow). Most of the curve was completely blocked by the contigs (yellow) curve. To show that the polished contigs (green) share nearly the same curve, the line was plotted more thickly so it can just barely be seen. Similarly, the Hi-C + RNA-seq scaffolds (purple) curve is very similar to the Hi-C only scaffolds (blue) curve. In this case, differences are more apparent. In certain places, e.g., at the highest peak, the Hi-C + RNA-seq scaffolds (purple) are plotted more thickly so it can be seen behind the Hi-C only scaffolds (blue).

## Transcriptome assembly

The transcriptome assembly generated by Trinity comprised 455K sequences with a mean sequence length of 1,177 bp. The N50 and L50 were 2.6 Kbp and 56K, respectively. The N90 and L90 were, respectively, 410 bp and 270K. Of the 3,640 single-copy orthologs in the ODB10 *Actinopterygii* set, 86.4% (3,144) were complete; 39.5% (1,241) of which were present only once in the transcript set. One hundred and twenty-eight (3.5%) single-copy orthologs were fragmented in the transcript set, 368 (10.1%) were missing (see Table 2).

## Computational genome annotation

Computational structural and functional annotation yielded 28.3 K protein-coding genes. Of these, 17.2 K and 15.6 K have annotated 5′ and 3′ untranslated regions (UTRs), respectively. Eighteen hundred (1,800) tRNA genes were also identified. The annotations are available with the assembly on GenBank. Approximately 6.5% of the genome comprised repetitive elements, and a summary of repeat types is available in Table 7.

**Table 7.** Summary of repeats.

|  | Copies | Length (Mbp) | Percent (%) of sequence |
|---|---|---|---|
| **Interspersed repeats** | **281,290** | **41.0** | **3.9** |
| SINE | 8,412 | 0.5 | 0.0 |
| Penelope | 1,919 | 0.4 | 0.0 |
| LINE | 60,890 | 8.4 | 0.8 |
| LTR | 6,692 | 2.6 | 0.3 |
| DNA transposon | 109,377 | 16.1 | 1.5 |
| Unclassified | 94,000 | 13.5 | 1.3 |
| **Tandem repeats** | **528,549** | **24.5** | **2.3** |
| Satellite | 1,306 | 0.1 | 0.0 |
| SSR | 464,086 | 21.0 | 2.0 |
| Low complexity | 63,157 | 3.4 | 0.3 |
| **Rolling-circles** | **21,964** | **1.7** | **1.2** |
| **Small RNA** | **4,191** | **0.7** | **0.1** |
| **Total** | **835,994** | **67.8** | **6.5** |

Summary of repeat content in the *Albula glossodonta* genome assembly as reported by RepeatMasker [93] using the Dfam v3.3 [94] and RepBase RepeatMasker v20181026 [95, 96] repeat libraries. Mbp: megabase pair; LINE: long interspersed nuclear element; LTR: long terminal repeat; SINE: short interspersed nuclear element; SSR: single sequence repeat.

**Table 8.** Pairwise $F_{ST}$ comparisons by island group.

|  | Amirantes | Farquhar | Aldabra |
|---|---|---|---|
| Farquhar | 0.0014* |  |  |
| Aldabra | 0.0005 | 0.0020* |  |
| Mauritius | 0.0034* | 0.0043* | 0.0040* |

Significance ($p < 0.05$) is indicated with an asterisk.

## Population genomic analysis

Cross-entropy scores generated by the model-based population differentiation analysis, sNMF, provided support for a single population of *A. glossodonta* across all localities. However, individual ancestry plots generated from sNMF results showed evidence of genetic differentiation in individuals from Mauritius (St. Brandon's Atoll), compared to the Seychelles sites (Figure 9A). This differentiation was corroborated by PCA visualization of the first two principal components, where St. Brandon's Atoll individuals clustered separately from the four Seychelles island groups (Figure 9B). Together, both population differentiation analyses indicated weak geographic population structure across all sampling localities, with reduced gene flow between St. Brandon's Atoll and the Seychelles sites.

   Pairwise $F_{ST}$ results also indicated greater genetic differentiation between St. Brandon's Atoll and all other island groups (Table 8). Estimates of observed and expected heterozygosity were similar across island groups (Table 9), suggesting no differences in genetic diversity between sampling localities and providing no evidence for distinguishing metapopulation processes such as inbreeding. A test of isolation by distance between sampling sites was not significant ($p = 0.1501$).

## Discussion

*Albula glossodonta* is an important fishery species in the Indo-Pacific for both subsistence and recreational purposes [21, 31, 115, 116]. Given this species' current "Vulnerable" IUCN status [7, 117] amid recent taxonomic uncertainties [4], understanding patterns of gene flow and population structure in *A. glossodonta* is important for fisheries management [31, 118].

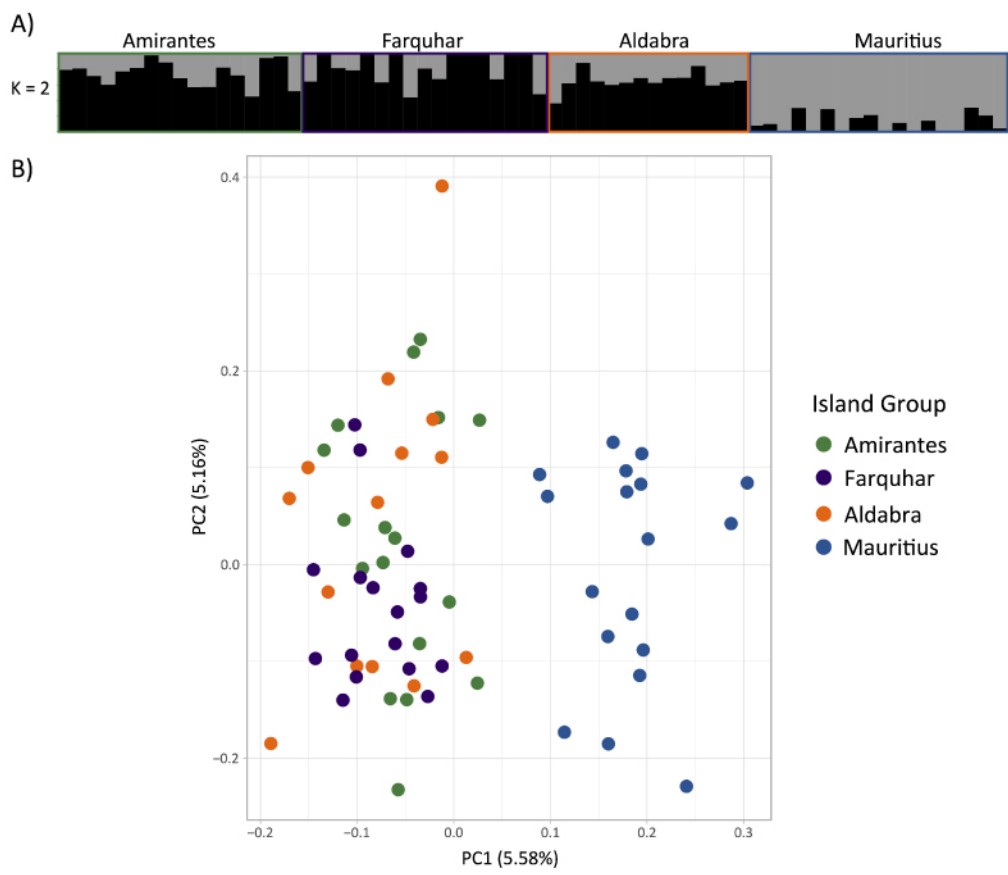

**Figure 9.** Population differentiation analyses. Weak geographic population structure is present across all sampling localities, with reduced gene flow between St. Brandon's Atoll and the Seychelles sites. Island groups are colored as in Figure 5. (A) Individual ancestry plots generated using sNMF, indicating $K = 2$ putative populations. (B) Principal component analysis biplot showing the first two principal components.

**Table 9.** Observed heterozygosity ($H_O$) and expected heterozygosity ($H_S$) for each island group.

| Island Group | $H_O$ | $H_S$ |
|---|---|---|
| Amirantes | 0.2800 | 0.2915 |
| Farquhar | 0.2901 | 0.2946 |
| Aldabra | 0.2589 | 0.2862 |
| Mauritius | 0.2829 | 0.2923 |

This annotated genome assembly is the first published for *A. glossodonta*, and it will provide a valuable resource for future studies of this vulnerable species. This genome is also the first for any bonefish species. It thus provides a basis for future genetic differentiation between all albulid species, especially for those in the same species complex (the *A. vulpes* complex) or that share the same range, i.e., primarily species in the *A. argentea* species complex. This will help future studies assess the population structure of bonefishes in the Indo-Pacific, enabling more informed conservation and fisheries management plans.

We observed a genetically homogenous population of *A. glossodonta* across five island atolls in the Seychelles Archipelago, with limited gene flow between Seychelles and Mauritius. Unlike highly migratory species such as eels (Anguillidae), which are close

relatives of bonefishes, adult bonefishes are known for high site fidelity with relatively short migrations (~10–100 km) [116, 119, 120]. We hypothesized that adult bonefishes would not migrate between the Seychelles islands, or between the Seychelles and St. Brandon's Atoll in Mauritius, since these distances span 400–2,000 km. Consequently, the observed trend of genetic homogeneity across the Seychelles is probably not a result of adult long-distance migrations, but pelagic larval dispersal, the primary dispersal mechanism for bonefishes [33, 121–123]. Bonefish larvae, also referred to as leptocephali, have a long pelagic larval duration ranging from 41–72 days, which enables them to drift long distances with ocean currents [22, 123]. The estimated pelagic larval duration for *A. glossodonta* is 57 days, based on observations of individuals from French Polynesia in the South Pacific [22]. The Seychelles islands are in the South Equatorial Current, which flows westwards from the Indian Ocean towards the eastern coast of continental Africa, enabling larvae to be transported across the Seychelles islands, even across depths exceeding 4,000 m (Figure 5) [124, 125].

Genetic homogeneity is not always an outcome of long pelagic larval duration, as demonstrated by *Anguilla marmorata*, for which 2–5 stocks were identified in the Indo-Pacific [126, 127], and *A. glossodonta*, where putative stocks between the Indian and Pacific Oceans were suggested [118]. Indeed, we found evidence of restricted gene flow between the Seychelles sampling sites and St. Brandon's Atoll, Mauritius, which is ~1,500–2,000 km from the Seychelles Islands (Figure 5). This genetic structuring was unexpected, given the long pelagic larval duration of *A. glossodonta*. However, there is evidence of limited gene flow between Seychelles and Mauritius in other marine fish species with pelagic larvae, such as *Lutjanid kasmira* [128], *Lethrinus nebulosus* [129], and *Pristipomoides filamentous* [130].

We attribute the observed genetic structure between Seychelles and St. Brandon's Atoll to the ocean currents in the SWIO and their role in larval transport [131, 132]. St. Brandon's Atoll is in the direct path of one of the bifurcated arms of the South Equatorial Current as it passes through the Mascarene Plateau [124, 133]. The South Equatorial Current pushes water westward, which may create a barrier to gene flow to islands south of Seychelles such as Mauritius and Réunion [129, 130, 133]. Although there are currently no bonefish – or even elopomorph – larval dispersal models for the Indian Ocean, pelagic larval dispersal simulation models of coral species in the SWIO corroborate the biogeographic break between Seychelles and Mauritius, suggesting connectivity is limited even when the pelagic larval duration is between 50–60 days [124, 133]. However, these models considered coral larvae, which are completely reliant on currents for their dispersal [124, 133, 134]. While the dispersal behavior of *A. glossodonta* larvae is unknown, we speculate that, similar to eels (Anguillidae; which also have long pelagic larval durations), bonefishes could disperse greater distances than passive corals by having the ability to swim (e.g., *Anguilla japonica* [135]) or may even take part in vertical migrations (e.g., *Anguilla japonica* [136, 137]). While officially undescribed, swimming ability in bonefish leptocephali has been observed [138], and vertical migrations have previously been theorized [121, 139].

Genome-wide datasets have enabled researchers to better delineate population connectivity across seascapes for marine species where conventional markers (e.g., mtDNA, microsatellites) have not provided sufficient genomic resolution [126, 140, 141]. Such advances in genomic sequencing have altered our view of population connectivity in other marine fishes such as yellowfin tuna (*Thunnus albacores* [142]) and the American eel



(*Anguilla rostrata* [143]). These studies, including ours, highlight the power of large genomic datasets for investigating connectivity in open ocean environments containing few, if any, natural barriers that were traditionally thought to drive population structure. Although there has been a rapid increase in the number of studies using next-generation sequencing datasets for marine fishes, few studies to date have employed the use of genomic datasets on elopomorphs, and none on bonefish [143–145].

## REUSE POTENTIAL

This is the first genome assembly and annotation for an albulid species, as well as the first use of a genome-wide SNP dataset to investigate population structure for *Albula glossodonta* or any bonefish species in the Indian Ocean. Individuals of *A. glossodonta* were genetically homogenous across four coralline island groups in the Seychelles Archipelago, but they showed evidence of genetic differentiation between the Seychelles and Mauritius (St. Brandon's Atoll). These patterns of connectivity are probably facilitated by pelagic larval dispersal, which is presumed to be strongly shaped by currents in the SWIO. Only with high-resolution genomic data were we able to discern this pattern of population structure between Seychelles and Mauritius. Our dataset serves as a valuable resource for future genomic studies of bonefishes to facilitate their management and conservation.

## DATA AVAILABILITY

The raw reads, genome assembly, and annotations are available under BioProject PRJNA668352 and BioSamples SAMN16516506–SAMN16516510 and SAMN17284271. The ddRAD reads are available under BioProject PRJNA702254, BioSamples SAMN18012541–SAMN18012606. The full, neutral, and outlier ddRAD-derived variant datasets, including annotations, are available from GigaDB [37].

## DECLARATIONS
## LIST OF ABBREVIATIONS

auNG: area under the NG curve; bp: base pair; BUSCO: Benchmarking Universal Single-Copy Orthologs; BYU: Brigham Young University; CLR: continuous long read; ddRAD: double digest restricted site-associated DNA; DNA: deoxyribonucleic acid; DNASC: DNA Sequencing Center; EDTA: ethylenediaminetetraacetic acid; EtOH: ethanol alcohol; Gbp: gigabase pair; HWE: Hardy–Weinberg equilibrium; IUCN: International Union for the Conservation of Nature; Kbp: kilobase pair; LINE: long interspersed nuclear element; LTR: long terminal repeat; Mbp: megabase pair; NCBI: National Center for Biotechnology Information; PacBio: Pacific Biosciences; PCA: principal component analysis; PCR: polymerase chain reaction; PE: paired-end; qPCR: quantitative polymerase chain reaction; RAM: random access memory; RNA: ribonucleic acid; RNA-seq: RNA sequencing; SMRT: Single-molecule, real time sequencing; SINE: short interspersed nuclear element; sNMF: sparse non-negative matrix factorization; SNP: single nucleotide polymorphism; SSR: single sequence repeat; SWIO: southwestern Indian Ocean; UTR: untranslated region; WGS: whole genome sequencing.

## ETHICAL APPROVAL

Not applicable.

## CONSENT FOR PUBLICATION

Not applicable.

## COMPETING INTERESTS

The authors declare that they have no competing interests.

## FUNDING

BDP was supported by a Conservation Scholarship from Fly Fishers International. ST was supported by the South African Institute for Aquatic Biodiversity, the Mandela Rhodes Foundation, the Marine Research Grant from the Western Indian Ocean Marine Science Association, and the Yale University Department of Ecology and Evolutionary Biology.

## AUTHORS' CONTRIBUTIONS

PDC: Conceptualization; Funding Acquisition; Investigation; Supervision; Resources; Writing - Review & Editing. DE: Methodology; Validation; Writing - Original Draft Preparation; Writing - Review & Editing. JRG: Conceptualization; Formal Analysis; Investigation; Supervision; Methodology; Visualization; Writing - Original Draft Preparation; Writing - Review & Editing. TPJ: Visualization. JSKK: Conceptualization; Funding Acquisition; Investigation; Supervision; Resources; Writing - Review & Editing. BDP: Conceptualization; Data Curation; Formal Analysis; Investigation; Methodology; Software; Visualization; Writing - Original Draft Preparation; Writing - Review & Editing. PGR: Funding Acquisition; Supervision; Resources; Writing - Review & Editing. ST: Investigation; Resources; Writing - Original Draft Preparation; Writing - Review & Editing.

## ACKNOWLEDGEMENTS

We thank the Brigham Young University DNA Sequencing Center (https://dnasc.byu.edu) and Office of Research Computing (https://rc.byu.edu) for their continued support of our research. We thank Elizabeth M. Wallace, Clayton Ching, Josiah Ching, Derek Olthuis, Zachary Emig, Weston Gleave, and the fly fishing guides from FlyCastaway (https://www.flycastaway.com) and Alphonse Fishing Company (https://www.alphonsefishingco.com), especially Daniel Hoenings and Matthieu Cosson, for the collection of samples in Hawai'i and the western Indian Ocean. We are grateful to Taryn Bodill and Martinus Scheepers of the South African Institute for Aquatic Biodiversity (https://www.saiab.ac.za) for laboratory assistance and Thomas Near of Yale University (https://www.yale.edu) for the use of laboratory space, funding, and equipment. We also thank the Seychelles Fishing Authority (http://www.sfa.sc), the Island Conservation Society (http://www.islandconservationseychelles.com), the Islands Development Company Ltd. (https://www.idcseychelles.com), the Seychelles Islands Foundation (https://www.sif.sc), the Ministry of Agriculture, Climate Change and Environment (https://pcusey.sc/about-meecc), and Shane and Hafiza Talma for their logistical support.

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
