## [Reviewer Report]

Reviewer name and names of any other individual's who aided in reviewer Changxu TianDo you understand and agree to our policy of having open and named reviews, and having your review included with the published papers. (If no, please inform the editor that you cannot review this manuscript.)YesIs the language of sufficient quality?YesPlease add additional comments on language quality to clarify if needed
Are all data available and do they match the descriptions in the paper? YesAdditional CommentsAre the data and metadata consistent with relevant minimum information or reporting standards? See GigaDB checklists for examples <a href="http://gigadb.org/site/guide" target="_blank">http://gigadb.org/site/guide</a>YesAdditional CommentsIs the data acquisition clear, complete and methodologically sound?YesAdditional CommentsIs there sufficient detail in the methods and data-processing steps to allow reproduction?YesAdditional CommentsIs there sufficient data validation and statistical analyses of data quality? YesAdditional CommentsIs the validation suitable for this type of data?YesAdditional CommentsIs there sufficient information for others to reuse this dataset or integrate it with other data?YesAdditional CommentsIn this paper, a high-quality genome of the Roundjaw Bonefish was successfully constructed, and population structure for Albula glossodonta or any bonefish species were well investigated with high-resolution genomic data. It serves as a valuable resource for future genomic studies of bonefishes to facilitate their management and conservation. Authors have presented the data in a meaningful way, I recommend the manuscript is publishable upon the following minor concerns are well addressed: 1. In the Tissue Collection and Preservation, why not use the same individual sample to complete DNA sequencing, but use the heart tissue of another individual for long-read sequencing and Hi-C sequencing. 2. In the Illumina RNA of Read Error Correction, why use the original read sequenced not filtered? 3. In the discussion section, it is suggested to add a discussion on the genomic results of this species.Any Additional Overall Comments to the Author
RecommendationMinor Revision

---

## [Reviewer Report]

Reviewer name and names of any other individual's who aided in reviewer Shengyong XuDo you understand and agree to our policy of having open and named reviews, and having your review included with the published papers. (If no, please inform the editor that you cannot review this manuscript.)YesIs the language of sufficient quality?YesPlease add additional comments on language quality to clarify if needed
Are all data available and do they match the descriptions in the paper? YesAdditional CommentsAre the data and metadata consistent with relevant minimum information or reporting standards? See GigaDB checklists for examples <a href="http://gigadb.org/site/guide" target="_blank">http://gigadb.org/site/guide</a>YesAdditional CommentsIs the data acquisition clear, complete and methodologically sound?YesAdditional CommentsIs there sufficient detail in the methods and data-processing steps to allow reproduction?YesAdditional CommentsIs there sufficient data validation and statistical analyses of data quality? YesAdditional CommentsIs the validation suitable for this type of data?YesAdditional CommentsIs there sufficient information for others to reuse this dataset or integrate it with other data?YesAdditional CommentsAny Additional Overall Comments to the AuthorIn the present study, the authors reported the genome assembly of bonefish Albula glossodonta, as well as population genomic analyses using ddRAD-seq. These genomic data should be useful for management and conservation of this species. Some comments are as follows. 1. The authors should show us the line numbers in their manuscript. 2. In Abstract and Result, the authors should provide fundamental genomic information such as genome size, heterozygosity ratio and repeat ratio, so we can have a better understanding of Albula glossodonta genome. 3. Also, the authors should provide the information of final genome assembly of this fish species, i.e. total length of genome assembly, the number and N50 of scaffolds, and among others. 4. What’s the meaning of NG50, LG50, and auNG in the manuscript? And what’s the difference between NG50 and N50? The authors should interpret why using these statistical data in the description of genome assembly part. 5. With an annotated genome assembly as reference, I suggested the identified SNPs should be annotated using SNPEff or annovar softwares. 6. Population genomic approach can uncover population divergence at a fine spatial scale. In this manuscript, relative high levels of genetic differentiation were detected between Mauritius and other three groups based on neutral SNP dataset, suggesting possible local adaptation in Mauritius population. I suggest the authors can further analyze population structure by using outlier dataset to reveal the influence of local adaptation on population differentiation.
RecommendationMajor Revision